# Novel Fermentates Can Enhance Key Immune Responses Associated with Viral Immunity

**DOI:** 10.3390/nu16081212

**Published:** 2024-04-19

**Authors:** Dearbhla Finnegan, Monica A. Mechoud, Jamie A. FitzGerald, Tom Beresford, Harsh Mathur, Paul D. Cotter, Christine Loscher

**Affiliations:** 1Food for Health Ireland, Science Centre South (S2.79), University College Dublin, Dublin 4, Ireland; dearbhla.finnegan7@mail.dcu.ie (D.F.); monica.mechoud@teagasc.ie (M.A.M.); jamie.fitzgerald@ucd.ie (J.A.F.); tom.beresford@teagasc.ie (T.B.); harsh.mathur@teagasc.ie (H.M.); paul.cotter@teagasc.ie (P.D.C.); 2School of Biotechnology, Faculty of Science, Glasnevin Campus, Dublin City University, D09 DX63 Dublin, Ireland; 3Teagasc Food Research Centre, Moorepark, Fermoy, P61 C996 Co. Cork, Ireland; 4College of Health and Agricultural Sciences, School of Medicine, University College Dublin, D04 V1W8 Dublin, Ireland; 5APC Microbiome Ireland, Biosciences Institute, Biosciences Research Institute, University College Cork, T12 R229 Cork, Ireland; 6VistaMilk, Teagasc, Moorepark, Shanacloon, Fermoy, P61 C996 Co. Cork, Ireland

**Keywords:** fermentates, functional food, immune boosting, immunomodulation, macrophage, viral immunity

## Abstract

Fermented foods have long been known to have immunomodulatory capabilities, and fermentates derived from the lactic acid bacteria of dairy products can modulate the immune system. We have used skimmed milk powder to generate novel fermentates using *Lb. helveticus* strains SC234 and SC232 and we demonstrate here that these fermentates can enhance key immune mechanisms that are critical to the immune response to viruses. We show that our novel fermentates, SC234 and SC232, can positively impact on cytokine and chemokine secretion, nitric oxide (NO) production, cell surface marker expression, and phagocytosis in macrophage models. We demonstrate that the fermentates SC234 and SC232 increase the secretion of cytokines IL-1β, IL-6, TNF-α, IL-27, and IL-10; promote an M1 pro-inflammatory phenotype for viral immunity via NO induction; decrease chemokine expression of Monocyte Chemoattractant Protein (MCP); increase cell surface marker expression; and enhance phagocytosis in comparison to their starting material. These data suggest that these novel fermentates have potential as novel functional food ingredients for the treatment, management, and control of viral infection.

## 1. Introduction

The term “fermentates” generally refers to “a powdered preparation, derived from a fermented [food] product and which can contain the fermenting microorganisms, components of these microorganisms, culture supernatants, fermented substrates, and a range of metabolites and bioactive components” [1]. In recent years, such fermented food products have been of ever-increasing interest, as they can exhibit health benefits including protection against infectious agents, immunomodulatory effects, anti-allergenic effects, anti-obesity effects, anti-oxidant effects and anti-anxiety effects [1]. Lactic acid bacteria (LAB), including *Lactobacilli* and *Bifidobacteria*, are responsible for the fermentation process within fermented foods, are generally regarded as safe (GRAS), and thus can be used in the production of functional foods [2]. Different strains of LAB can produce different fermentation products that can interact with microorganisms during intestinal transit and have the ability to therefore interact with the cells of the intestinal wall [3]. In the generation of these fermentates, the LAB undergo a heat killing phase, creating a fermentate or postbiotic ingredient that has bioactivity associated with the secondary metabolites present, as opposed to a viable LAB strain. Postbiotics are ideal components for the development of a large range of novel health-promoting consumable products, as functional foods and potential nutraceuticals [4].

With recent viral outbreaks like that of the SARS-CoV-2 virus, Monkeypox virus, and most recently the Langya virus, as well as the yearly outbreaks of seasonal influenza, there is a need to explore new ways of enhancing viral immunity [5]. This study is one of the first to explore the effects of novel milk fermentates, derived from *Lb. helveticus*, to impact on immune mechanisms that are critical to viral immunity. The objective of this study was to examine the effects of two milk fermentates made using *Lb. helveticus* SC234 or SC232 (sourced from Lallemand, Quebec, CA) on murine macrophage cells challenged with the viral immune stimulus Loxoribine (LOX), or an inflammatory immune stimulus lipopolysaccharide (LPS; *Escherichia coli* 055:B5). The effects of these novel fermentates on cell viability, cytokine secretion interleukin-1β (IL-1β), IL-6, IL-10, tumour necrosis factor (TNF)-α, IL-12p40, IL-27, nitric oxide production (NO) and arginase activity in M1- and M2-polarised macrophages, chemokine secretion (MIP)-1, MIP-2, Monocyte Chemoattractant Protein (MCP), cell surface marker expression (major histocompatibility complex (MHC) II, CD86, Toll-like receptor ligand (TLR) 4, cluster-differentiated (CD) 80, CD14, TLR2, CD40, and MHCI), and phagocytosis were investigated in LOX- and LPS-activated murine-derived macrophage. We demonstrate that fermentates SC234 and SC232 increase the secretion of cytokines IL-1β, IL-6, TNF-α, IL-27, and IL-10; promote an M1 pro-inflammatory phenotype for viral immunity via NO induction; decrease chemokine expression of MCP; increase cell surface marker expression; and enhance phagocytosis in comparison to their starting material.

## 2. Materials and Methods

### 2.1. Generation of Dairy-Based Fermentates

Skim Milk Powder (SMP) was used as a substrate for the generation of the fermentates used in this study. SMP was reconstituted at 10% *w*/*v* in distilled water to generate Reconstituted Skim Milk (RSM), autoclaved, cooled, and stored at 4 °C for a maximum of 7 days. An inoculum of frozen mother culture stocks of the individual strains *Lb. helveticus* SC232 and *Lb. helveticus* Lafti L10 SC234 (which were previously prepared in 10% *w*/*v* RSM) was added to 10% RSM and incubated for 24 h at 37 °C under aerobic conditions without agitation, to generate these individual fermentates derived from the above-mentioned individual strains. From these cultures, a further inoculum was added to 10% *w*/*v* RSM and incubated for 24 h at 37 °C under aerobic conditions again without agitation. These fermentates were subjected to a heat treatment step to generate the fermentates which contained one of the heat-killed LAB strains mentioned above. After cooling to room temperature, the pH of the fermentates was neutralized. These fermentates were aliquoted and immediately frozen at −80 °C until further analysis. Non-fermented RSM samples subjected to the same heat-treatment mentioned above were used as negative controls for all experiments described herein.

### 2.2. Cell Culture

J774.A.1 Murine Macrophage, purchased from the European Collection of Animal Cell Cultures (Salisbury, UK), were maintained in Dulbecco Modified Eagle Medium (DMEM) supplemented with 10% heat-inactivated foetal bovine serum (FBS), and 1% Penicillin–Streptomycin Antibiotic obtained from Bioscienes (Dublin, Ireland), and incubated at 37 °C, with 5% CO_2_ and 95% humidified air. Cells were passaged every three to four days at a confluence of 80–90%. Cells were sub-cultured at a ratio of 1:10. Bone Marrow-Derived Macrophage cells (BMDMs), harvested from the bone marrow of 6–8-week-old female BALB/c mice obtained from Charles River (Margate, UK), were cultured in complete Roswell Park Memorial Institute (RPMI) 1640 medium, containing 25 ng/mL rM-CSF (Merck, Haverhill, UK).

### 2.3. Cell Viability

Cell viability was determined using the CellTiter 96^®^ AQueous One Solution Cell Proliferation Assay and conducted as per the manufacturer’s instructions (MyBio, Kilkenny, Ireland). Macrophages were seeded at a concentration of 1 × 10^6^ cells/mL in a flat-bottom 96-well plate, and incubated for 24 h at 37 °C in a 95% humidified air and 5% CO_2_ atmosphere. Cells were treated with 25 mg/mL of the fermentate sample for 1 h and incubated under the same conditions, before stimulation with LOX 0.5 mM and LPS 100 ng/mL for 24 h. DMSO was included as a positive control to induce cell death. After 24 h, 20 μL of the thawed CellTiter96^®^ Aqueous One Solution was added to each well of the 96-well plate, and incubated at 37 °C for 3 h in a humidified 5% CO_2_ atmosphere. Absorbance was read at 490 nm using Versamax^TM^ 96-well plate reader (VWR, Dublin, Ireland). Cell viability was expressed as the percentage viability of the treatment group relative to the control group.

### 2.4. Enzyme-Linked ImmunoSorbent Assay (ELISA)

Determination of the effects of the fermentate samples on cytokine and chemokine production in the activated macrophages required harvesting of the cell supernatants, and subsequent analysis using commercial DuoSet ELISA kits (R&D Systems Europe, Abdingdon, Oxon, UK), according to the manufacturer’s instructions. This allowed for the quantification of the cytokines IL-1β, IL-6, IL-10, TNF-α, IL-12p40, and IL-27, as well as the chemokines MCP, MIP-1, and MIP-2.

### 2.5. Nitric Oxide (NO) and Arginase Assay

NO production was determined by measuring the NO_2_^−^ in the cell supernatants of the cultured macrophage via a Griess assay, carried out as per manufacturer’s instructions (MyBio, Kilkenny, Ireland). Cell lysates were prepared and analysed for arginase activity via the proportional detection of urea, a direct result of arginase catalysing the conversion of arginine to urea and ornithine, using a commercial kit and following the manufacturer’s instructions (Merck, Haverhill, UK). BMDM cells were seeded at a concentration of 5 × 10^5^ cells per well in 24-well plates and incubated for 30 min at 37 °C. BMDMs were stimulated with 25 mg/mL fermentates for 3 h and incubated under the same conditions. BMDM cells were polarised towards M1 macrophages by adding LPS (100 ng/mL) in the presence of 20 ng/mL rIFN-γ or towards M2 cells by adding 20 ng/mL rIL-4, 20 ng/mL IL-13, and 20 ng/mL rTGF-B and incubated for 24 h at 37 °C. After 24 h, supernatant was harvested and Griess assay was carried out to quantify the NO_2_^−^ present, while an arginase assay was carried out to determine the arginase activity within the cell.

### 2.6. Cell Surface Marker Expression Analysis

The determination of cell surface markers present on the J774.A.1 macrophage was carried out via cell surface marker staining using the fluorescently labelled antibodies FITC, APC, and PE. J774.A.1 macrophages were seeded at a concentration of 1 × 10^6^ cells/mL in a 6-well plate, stimulated with appropriate treatments and incubated at 37 °C, with 5% CO_2_ and 95% humidified air. Cells were blocked with FBS for 15 min, before being harvested via centrifugation at 2000 rpm for 5 min at 4 °C. Cells were resuspended in FACS buffer. Cell suspension was plated in a 96-well round-bottom plate and centrifuged. Supernatant was aspirated, and cells resuspended in 1:1000 dilutions of antibodies (FITC, APC, PE) and incubated for 30 min at 4 °C. Cells were centrifuged and washed 3 times in FACS buffer. Cells were resuspended in FACS buffer and transferred to FACS loading tubes. Cells were analysed using a BD FACSAria 1 system flow cytometer. Raw FCS files were analysed, and data were graphed using V10.0 FlowJo software. Cell surface marker expression was determined for cell surface markers MHCI, MHCII, TLR2, TLR4, CD40, CD14, CD80, and CD86.

### 2.7. Phagocytosis Assay

J774A.1 macrophages were seeded at 1 × 10^6^ cells/mL in 6-well plates and incubated overnight at 37 °C in a humidified, 5% CO_2_ atmosphere. Cells were stimulated with sample for 1 h, incubated under the same conditions. Subsequently, cells were stimulated with 100 ng/mL LPS and LOX for 4 h. Cells were incubated with 1 µm fluorescent latex beads (Merck, Haverhill, UK) at a concentration of 20 beads per cell for 1 h at 37 °C in a humidified 5% CO_2_ atmosphere. Cells were scraped from the cell culture plate and pelleted via centrifugation at 4 °C at 2000 rpm for 5 min. Cells were resuspended and washed twice in 1 mL FACs buffer via centrifugation. Cells were resuspended in FACs buffer and added to FACs tubes. The uptake of beads was measured by flow cytometry on a FACSAria™ flow cytometer. Data were analysed using FlowJo software (Treestar, Woodburn, OR, USA). MFI and percentage of phagocytosing cells were the two outputs measured.

Data were represented as the MPI. This is a representation to incorporate the MFI from the phagocytosed beads as well as the percentage of phagocytosing cells in the viable population of cells and compare them to the control, which represents baseline phagocytosis. The MPI is calculated as follows:$$\mathrm{MPI}=\frac{\%\;phagocytes\times MFI}{control\;phagocytes\times control\;MFI}$$

### 2.8. Statistical Analysis

Statistical analysis was carried out using a one-way ANOVA to compare variance among the means of different sample groups. A Newman–Keuls post-test was used to determine significance among the samples. The level of statistical significance was indicated by * (*p* < 0.05), ** (*p* < 0.01), and *** (*p* < 0.001).

### 2.9. Ethical Statement

The care, treatment, and experiments involved in this study were approved by the Research Ethics Committee (REC) of Dublin City University (Approval ID: DCUREC/2011/008).

## 3. Results

### 3.1. Immune-Boosting Effects of Fermentates on Cytokine Secretion

Initially, in the preliminary experiments carried out by the laboratory, an MTS assay confirmed that the fermentates SC232 and SC234 in the presence/absence of LOX or LPS had no effect on the viability of either J774.A.1 cells and BMDMs.

Additionally, an ELISA was performed on the cell line J774.A.1 macrophage to assess the bioactivity of fermentates SC232 and SC234 in the presence/absence of LOX or LPS. The novel fermentates altered the secretion of cytokines in response to LOX and LPS when compared to the respective controls in J774.A.1s. With this knowledge, we were then able to carry out the study in question.

In order to confirm the effects of our fermentates in macrophages, we assessed their effects in primary cells. SC234 and SC232 significantly affected the secretion of cytokines in response to LOX and LPS when compared to the respective controls in BMDMs (Figure 1). IL-1β (*p* < 0.001), IL-6 (*p* < 0.001), IL-12p40 (*p* < 0.033), IL-10 (*p* < 0.033), and TNF-α (*p* < 0.001) were increased in the presence of LOX with only low levels of IL-27 detected. IL-6 (*p* < 0.001), TNF-α (*p* < 0.001), and IL-27 (*p* < 0.001) were increased in the presence of LPS with only low levels of IL-1β, IL-10, and IL-12p40.

In the presence of LOX, SC234 significantly enhanced IL-6, and IL-10 secretion (*p* < 0.002; *p* < 0.001), but decreased IL-1β secretion (*p* < 0.001), with no significant effects on the other cytokines measured when compared to control cells. In contrast, SC232 significantly increased IL-6 (*p* < 0.001), IL-12p40 (*p* < 0.001), and IL-10 (*p* < 0.001), with no significant effect on the other cytokines. The exposure of cells to SC234 in the presence of LPS resulted in an increase in IL-10 (*p* < 0.001), but a decrease in IL-12p40 (*p* < 0.002), with no significant effect on the other cytokines. SC232 in the presence of LPS resulted in an increase in IL-1β (*p* < 0.002), IL-6 (*p* < 0.002), IL-10 (*p* < 0.001), and TNF-α (*p* < 0.001) but no change in IL-12p40 or IL-27. Interestingly, the exposure of cells to SC234 alone, in the absence of either LOX or LPS stimulation, resulted in enhanced secretion of IL-10 (*p* < 0.001), but decreased IL-12p40 (*p* < 0.033) and TNF-α secretion (*p* < 0.002), and the exposure of cells to SC232 alone resulted in decreased secretion of IL-1β (*p* < 0.002).

The non-fermented RSM, used as a negative control, did affect cytokine secretion in the presence of LOX with an increase in IL-6 (*p* < 0.001), IL-12p40 (*p* < 0.001), and IL-27 (*p* < 0.001), but a decrease in IL-1β (*p* < 0.001). Furthermore, RSM enhanced IL-6 (*p* < 0.001), but decreased IL-12p40 (*p* < 0.033) in the presence of LPS. RSM, in the absence of TLR stimulation, increased TNF-α (*p* < 0.033).

Given that RSM itself had some effects, we also compared the fermentates to the RSM. In the absence of TLR stimulation, SC232 increased IL-10 (*p* < 0.002). In the presence of LOX, SC234 increased IL-10 (*p* < 0.001) but decreased IL-12p40 (*p* < 0.001) and IL-27 (*p* < 0.001) relative to the RSM. In the presence of LOX, SC232 increased IL-1β (*p* < 0.001) and IL-10 (*p* < 0.033), but decreased IL-12p40 (*p* < 0.033) and IL-27 (*p* < 0.001). In the presence of LPS, SC234 increased IL-10 (*p* < 0.001), but decreased IL-6 (*p* < 0.002), and SC232 increased IL-1β (*p* < 0.033) and IL-10 (*p* < 0.001), relative to the RSM.

Given that the fermentates had clear effects on macrophages, we sought to determine if they could exert specific effects on M1- and M2-polarised macrophages. In Figure 2, A shows the cytokine secretion profiles of M0-, M1-, and M2-polarised macrophages. M1 macrophages secrete high levels of IL-6 and TNF-α (*p* < 0.001; *p* < 0.001), while M2 macrophages secrete only small concentrations of IL-6 and TNF-α but secrete higher concentrations of IL-10 compared with the M0-unpolarised macrophages and the M1-polarised macrophages which secrete undetectable levels. The M0 macrophages secrete undetectable levels of IL-6, TNF-α, or IL-10.

SC234 increased IL-6 (*p* < 0.001), IL-10 (*p* < 0.001), and TNF-α (*p* < 0.001) in M0 BMDMs (Figure 2A–D). M0 BMDMs in the presence of SC232 showed increased IL-6 (*p* < 0.002) and IL-10 (*p* < 0.001) relative to the M0 control. In M1-polarised BMDMs, SC234 increased IL-6 (*p* < 0.001), IL-10 (*p* < 0.001), and TNF-α (*p* < 0.001), and SC232 increased IL-6 (*p* < 0.001), IL-10 (*p* < 0.001), and TNF-α (*p* < 0.002), relative to the M1 control. In M2-polarised BMDMs, SC234 increased IL-10 (*p* < 0.001) and TNF-α (*p* < 0.002), and SC232 increased IL-10 (*p* < 0.001), relative to the M2 control.

### 3.2. Immune-Boosting Effects of Fermentates on Nitric Oxide Production and Arginase Activity

Nitric oxide production and arginase activity are classical markers of M1 and M2 macrophages.

Figure 3A exhibits that M1 macrophages secreted high levels of NO (*p* < 0.001), while M0 and M2 macrophages secreted only small concentrations of NO.

Figure 3B shows that M2 macrophages have high levels of arginase activity (*p* < 0.033), while M0 and M1 macrophages have much lower levels of arginase activity.

Figure 3C demonstrates that SC234 and SC232 significantly increased NO production in M0 (*p* < 0.001; *p* < 0.001), M1 (*p* < 0.001; *p* < 0.001), and M2 macrophages (*p* < 0.001; *p* < 0.001). However, it was the M1 BMDMs in the presence of SC234 and SC232 that produced the highest concentration of NO.

Figure 3D exhibits that SC234 and SC232 increased arginase activity in M0 (*p* < 0.002; *p* < 0.033). In M1 BMDMs, only SC234 increased arginase activity (*p* < 0.033), and in M2 BMDMs there was no significant effect.

### 3.3. Immune-Boosting Effects of Fermentates on Chemokine Secretion

Figure 4 exhibits that our novel fermentates significantly affected the secretion of chemokines in response to LOX and LPS in BMDMs. MCP (*p* < 0.001), MIP-1 (*p* < 0.001), and MIP-2 (*p* < 0.001) were increased in the presence of LOX relative to LOX control. MCP (*p* < 0.001) and MIP-2 (*p* < 0.001) were increased in the presence of LPS, with only a small increase seen in MIP-1 relative to LPS control.

In the presence of LOX, SC234 and SC232 significantly decreased MCP (*p* < 0.033). Exposure of cells to SC234 in the presence of LPS resulted in an increase in MIP-1 (*p* < 0.001), but a decrease in MCP (*p* < 0.001), relative to LOX control. Similarly, SC232 in the presence of LPS resulted in an increase in MIP-1 (*p* < 0.002), but a decrease in MCP (*p* < 0.033), relative to LPS control. Interestingly, exposure of cells to SC234 alone, in the absence of either LOX or LPS stimulation, resulted in enhanced secretion of MIP-1 (*p* < 0.033) and MIP-2 (*p* < 0.001) relative to control cells. Exposure of cells to SC232 alone, in the absence of either LOX or LPS stimulation, resulted in increased secretion of MIP-2 (*p* < 0.001), relative to control cells.

The RSM control itself did affect chemokine secretion in the presence of LPS with an increase in MIP-1 (*p* < 0.001), but decreased MCP (*p* < 0.033), relative to LPS control. Furthermore, RSM in the absence of TLR stimulation enhanced MIP-2 (*p* < 0.033) relative to the control cells.

Given that RSM itself had some effects, we also compared the fermentates to RSM. In the presence of LOX, SC234 and SC232 decreased MCP (*p* < 0.033).

### 3.4. Immune-Boosting Effects of Fermentates on Cell Surface Marker Expression

Figure 5 exhibits that fermentates significantly affected the expression of cell surface markers in response to LOX and LPS. LOX significantly increased the expression of CD86, CD14, CD40, TLR4, and CD80 (*p* < 0.001), and LPS significantly increased the expression of CD86, CD14, CD40, TLR4, CD80, and MHCI (*p* < 0.001).

In the presence of LOX, SC234 further enhanced the expression of MHCII (*p* < 0.033), CD86 (*p* < 0.001), TLR2 (*p* < 0.001), MHCI (*p* < 0.001), CD14 (*p* < 0.001), CD40 (*p* < 0.001), and CD80 (*p* < 0.001). In the presence of LPS, SC234 further enhanced the expression of MHCII (*p* < 0.033), TLR4 (*p* < 0.033), CD86 (*p* < 0.001), TLR2 (*p* < 0.001), CD14 (*p* < 0.001), CD40 (*p* < 0.001), and CD80 (*p* < 0.001). In the absence of TLR, SC234 increased MHCII (*p* < 0.002), CD86 (*p* < 0.001), CD14 (*p* < 0.001), CD40 (*p* < 0.001), CD80 (*p* < 0.001), TLR2 (*p* < 0.001), and MHCI (*p* < 0.001).

In the presence of LOX, SC232 increased MHCII (*p* < 0.033) and TLR4 (*p* < 0.033) expression, but decreased CD14 (*p* < 0.002), CD40 (*p* < 0.033), and MHCI (*p* < 0.002) expression.

In the presence of LPS, R00352 further enhanced the expression of MHCII (*p* < 0.001), TLR4 (*p* < 0.001), CD14 (*p* < 0.001), CD86 (*p* < 0.001), CD80 (*p* < 0.001), TLR2 (*p* < 0.001), CD40 (*p* < 0.001), and MHCI (*p* < 0.033). In the absence of TLR, SC232 increased the expression of MHCII, CD14, CD80, TLR2, and MHCI (*p* < 0.001), and further enhanced TLR4, CD40, and CD86 (*p* < 0.001).

In the presence of LOX, RSM further enhanced the expression of MHCII (*p* < 0.001), TLR4 (*p* < 0.001), CD14 (*p* < 0.001), CD40 (*p* < 0.001), CD86 (*p* < 0.001), CD80 (*p* < 0.001), TLR2 (*p* < 0.001), and MHCI (*p* < 0.001). In the presence of LPS, RSM further enhanced the expression of MHCII (*p* < 0.001), TLR4 (*p* < 0.001), CD14 (*p* < 0.001), CD40 (*p* < 0.002), CD86 (*p* < 0.001), CD80 (*p* < 0.001), and TLR2 (*p* < 0.001). In the absence of TLR, RSM increased the expression of MHCII, TLR4, CD14, CD40, CD86, CD80, TLR2, and MHCI (*p* < 0.001).

Given that RSM itself had some effects, we also compared the fermentates to the RSM control. In the absence of TLR, SC234 decreased TLR4 and MHCI (*p* < 0.001; *p* < 0.002), but further increased CD14 (*p* < 0.001), CD40 (*p* < 0.001), CD80 (*p* < 0.001), and TLR2 (*p* < 0.033). In the absence of TLR, SC232 decreased TLR4 (*p* < 0.033) and CD86 (*p* < 0.033), but further increased CD40 (*p* < 0.033). In the presence of LOX, SC234 decreased TLR4, CD86, and TLR2 (*p* < 0.001). In the presence of LOX, SC232 decreased TLR4, CD14, CD40, CD86, CD80, TLR2, and MHCI (*p* < 0.001). In the presence of LPS, SC234 decreased TLR4 (*p* < 0.002) and CD86 (*p* < 0.001), but further increased CD40 (*p* < 0.001), CD80 (*p* < 0.002), and CD14 (*p* < 0.033). In the presence of LPS, SC232 decreased CD86 (*p* < 0.001), CD40 (*p* < 0.033), and MHCI (*p* < 0.033).

### 3.5. Stimulation of LOX and LPS Activated J774 with 25 mg/mL Fermentates Affect Phagocytosis

The procedure for phenotypic analysis of cell phagocytosis when stimulated with TLR ligands was carried out as previously described using 1 µm fluorescent FITC latex beads. MFI, percentage of phagocytosing cells, and MPI were measured.

#### 3.5.1. MFI

Figure 6A–C demonstrate that stimulation of J774 cells with LOX and LPS significantly increased the MFI (*p* < 0.033; *p* < 0.002) and that the addition of SC234 or SC232 to LOX- and LPS-stimulated cells suppressed this increased MFI. SC234 and SC232 alone had no effect. Figure 6C demonstrates that the presence of RSM had a similar effect on LOX* and LPS-stimulated cells to SC234 and SC232 and RSM alone had no effect.

#### 3.5.2. Percentage of Phagocytes

Figure 6D–F demonstrate that LOX and LPS increased the percentage of phagocytosing cells (*p* < 0.002; *p* < 0.001) which were suppressed by the presence of SC234 but not SC232. Interestingly, SC234 alone also increased the percentage of phagocytosing cells, and RSM alone or in the presence of LOX or LPS had no significant effect.

#### 3.5.3. MPI

Figure 6G–I demonstrates that stimulation of J774 cells with LOX and LPS significantly increased the MPI (*p* < 0.033; *p* < 0.001) which was suppressed by the addition of SC234. However, SC234 alone enhanced the MPI. The addition of SC232 to LOX-stimulated cells also suppressed the increased MPI, in contrast to the maintained MPI in LPS-activated cells. SC232 alone had no effect and RSM had a similar effect on LOX- and LPS-stimulated cells to SC234.

## 4. Discussion

This study demonstrates the potential of the novel fermentates, SC232 and SC234, in modulating key macrophage functions, which are central to the protection and clearance of viral infections. Macrophages act as scavengers, enabled by the presence of pattern recognition receptors, to alert the immune system through chemokine and cytokine secretion and antigen presentation, and to engulf and destroy invading pathogens via phagocytosis [6]. Macrophages play a critical role in both innate and viral immunity and so are an important cell to target to enhance their capabilities.

Initially, preliminary studies carried out by the laboratory on a large panel of fermentates used a dose range of 5 mg/mL, 10 mg/mL, 25 mg/mL, and 50 mg/mL fermentates to reveal 25 mg/mL as the optimal dose for fermentate bioactivity, and thus this dose of 25 mg/mL was used for such further in-depth analysis. Following confirmation that the fermentates and the starting substrate, RSM, did not affect cell viability, we demonstrated that cytokine secretion in J774.A.1 and BMDM macrophages are positively affected by the presence of SC232 and SC234 when compared to the effects of the RSM. Furthermore, these effects differ depending on the mode of activation of the cell. J774.A.1 and BMDM cells, when activated with LOX and LPS in the presence of SC232 and SC234, showed enhanced levels of secretion of IL-1β, IL-6, IL-27, and IL-10. BMDM cells, when activated with LOX and LPS in the presence of SC232 and SC234, showed enhanced levels of secretion of IL-12p40 and IL-27 following LOX exposure, but decreased IL-12p40 and IL-27 secretion following LPS exposure. This suggests that they may have specific effects on the immune system in the presence of a viral ligand.

In polarised BMDMs, SC232 and SC234 show a similar profile of activity. M0- and M1-polarised BMDMs in the presence of SC232 and SC234 secreted high levels of IL-6 and TNF-α, and M0-, M1-, and M2-polarised BMDMs in the presence of SC232 and SC234 secrete high levels of IL-10.

Given the importance of IL-6, TNF-α, IL-12p40, and IL-27 in aiding the immune system during viral infections such as influenza, vaccinia virus, HIV, and herpes simplex [7,8,9,10,11,12,13] and supporting viral immunity, a fermentate that can enhance these cytokines could be beneficial. In response to a viral activation, the novel fermentates SC234 and SC232 can enhance not only IL-6, IL-12p40, and IL-27 in BMDMs, but also support IL-6 and TNF-α secretion in polarised BMDMs; thus, they have potential to support viral immunity. These effects are not the same in the presence of LPS with decreased IL-12p40 and IL-27, and so the unique bioactivity we see in the fermentates’ ability to enhance cytokine response to viral ligands further supports their possible specificity in enhancing viral immunity.

Similarly, this anti-viral profile is seen in other classic markers of M1 and M2 macrophage, NO production, and arginase activity. M0, M1, and M2 BMDMs in the presence of SC232 and SC234 produce high levels of NO. M0 and M1 BMDMs in the presence of SC232 and SC234 promoted low levels of arginase activity. This emphasises the largely pro-inflammatory M1 profile of SC232 and SC234. The effect on NO production is of particular interest given that NO production is necessary for viral clearance via inducible NO synthase (iNOS), and depending on the virus can have direct antiviral properties, limiting the severity of virus-induced disease [14,15]. NO production is linked to the M1 killing/fighting phenotype, whereby arginine is metabolised via iNOS to NO and citrulline to aid M1 macrophages in the production of Th1 responses for fighting infection, and aiding in the drive and recruitment of pro-inflammatory cytokines useful for the defence of the immune system against viruses [16,17].

It has previously been demonstrated that an isolated acidic polysaccharide from the fungus *Cordyceps militaris* enhanced mRNA expression of IL-1β, IL-6, IL-10, and TNF-α, increased NO production, and induced iNOS mRNA and protein expression in RAW 264.7 macrophage cells, as well as increasing TNF-α and IFN-γ in mice, to decrease virus titres in the bronchoalveolar lavage fluid and the lungs of mice infected with influenza A virus to increase survival rate [18]. Similarly, a study involving *Lactobacillus helveticus* has showed a trending decrease in influenza-like illness in an elderly population, suggesting it could elicit a similar effect [19]. Another study showed that germinated *Rhynchosia nulubilis* fermented with *Pediococcus pentosaceus* SC11 has immune-enhancing and anti-viral effects, inhibiting 3CL protease activity in SARS-CoV in immunocompromised mice, increased T lymphocyte production and splenocyte proliferation, increased phagocytic activity, NO production via induction of iNOS, mRNA expression of IFN-γ, IFN-α, and ISG15 in RAW 264.7 macrophages, and subsequent increase in the expression of TNF-α [20], suggesting the role of GRC-SC11 in immunosuppressed patients for support against SARS-CoV. Similarly, *Grifola frondosa* extract can induce the expression of TNF-α mRNA in Madin–Darby canine kidney cells leading to the production of TNF-α, with subsequent inhibition of viral growth of influenza A/Aichi/2/68 virus [21]. TNF-α possesses anti-viral activity through its synergy with IFNs to induce resistance to DNA and RNA of viruses in diverse cell types, selectively killing the virus, necessary for the initiation and continuation of inflammation and immunity, adhesion molecule expression, and recruitment of leukocytes [22,23]. Other studies by Takeda et al. showed that LAB, in particular the strain *Lactiplantibacillus plantarum* 06CC2 from cow cheese, increased the production of IL-12 and IL-12p40 in vitro and in vivo [24]. *Lactiplantibacillus plantarum* 06CC2 has been associated with the enhancement of the Th1 response, and resulted in the alleviation of influenza virus infection in mice [25]. *Rapanea melanophloeos* has been shown to increase IL-27 production, ultimately increasing IL-10 production, to decrease the viral titre of influenza A virus in MDCK cells, suggesting its role as an anti-influenza treatment [26]. IL-27 activates and promotes the production of IFNs which are associated with various antiviral activities, support plasmacytoid DCs to sense viral DNA and RNA, promote macrophage differentiation and polarisation, increase TLR expression, and promote IL-10 cytokine production [7]. Therefore, not only is IL-27 important in viral immunity but IL-27 leads to subsequent enhanced IL-10 production for viral clearance, and thus where IL-27 is increased, IL-10 will often reflect this. IL-10 is a CD4-produced Th2 cytokine with the ability to indirectly suppress Th1 responses, downregulate the antigen-presenting capacities of APC, inhibit the activation and effector function of T cells, monocytes, and macrophages, therefore limiting host immune response to invading pathogens and ultimately preventing damage to the host from overactivation of the pro-inflammatory molecules, and provides a supportive role in effective virus clearance [27,28,29,30]. Additionally, in work carried out in our laboratory, we have further demonstrated the positive impact that fermentates SC232 and SC234 had on viral immunity [31]. SC232 and SC234 positively impacted on the secretion of the cytokines IL-6, TNF-α, IL-12p40, IL-23, IL-27, and IL-10, and decreased IL-1β in primary bone marrow-derived dendritic cells (BMDCs) stimulated with a viral ligand, thus further establishing their role as viral immune-boosting fermentates with positive effects in a range of immune cells [31].

Having established SC234 and SC232 as potential anti-viral fermentates that enhance viral immunity through their increased secretion of cytokines important for viral immune responses, we then extended our analysis to chemokines which are critical in supporting the immune system in response to viral infection and host protection. We demonstrate that chemokine secretion in BMDM macrophages is positively affected by the presence of SC232 and SC234 when compared to the activity of the RSM. Chemokines are critical in order to mediate macrophage chemotaxis, cell trafficking, and in the regulation of M1 and M2, and regulate differentiation of monocytes into dendritic cells (DCs), to attract macrophage towards the site of injury or infection [32,33,34]. MCP-1 (CCL2) is a key chemokine for the regulation of migration and infiltration of monocytes, playing a critical role in inflammation [33,35]. MIP-1 α (CCL3), on the other hand, plays a key role in viral immunity, being a chemotactic chemokine secreted by macrophages to aid in the recruitment of cells, wound healing, inhibition of stem cells, and maintaining of effector immune responses, and is a key mediator of virus-induced inflammation [36,37]. Similarly, MIP-2 (CXCL2) plays a role in viral immunity, aiding in neutrophil recruitment and activation, and is a potent chemoattractant secreted by macrophage and epithelial cells that plays a critical role in LPS-induced inflammation, as well as aiding in suppressing of viral replication [38,39,40]. Therefore, while MCP is important for inflammation, it is MIP-1 and MIP-2 that play critical roles in the context of viral infection, with roles in virus-induced inflammation and suppressing viral replication [33,36,37,38,39,40]. It is important, however, that these chemokines are not overexpressed, as this leads to pathogenesis of many inflammatory diseases including cancers and rheumatoid arthritis, and viruses such as coronavirus [34,36].

BMDMs, when activated with LOX in the presence of SC232 and SC234, decrease MCP. LOX-activated BMDMs in the presence of RSM, SC232, and SC234 maintain MIP-1 and MIP-2 concentrations. MCP, MIP-1, and MIP-2 are all enhanced in the presence of the fermentates alone. This again highlights the potential for SC232 and SC234 to be anti-viral, as they maintain concentrations of the viral-associated chemokines MIP-1 and MIP-2 and decrease MCP, which is often associated with pathogenesis of viral infections [34,36].

It is well established that decreases in MCP can be linked with the ability to inhibit viruses such as HIV [41]. The use of the Chinese herbal medicine Shikonin, from the dried root of *Lithospermum erythrorhizon*, has been linked to the ability to inhibit HIV-1 through its interactions inhibiting MCP and MIP-1 [41]. Lianhuaqingwen capsules from the traditional Chinese medicine prescriptions Maxing Shigan Tang and Yinqiao San decreased the expression of MCP-1, resulting in anti-viral activity for the treatment of influenza viral infection [42]. The similar activities of SC234 and SC232 on chemokines may highlight their anti-viral potential.

Next, we went on to assess the effect of the fermentates SC234 and SC232 on cell surface marker expression in J774.A.1. These cell surface proteins play a critical role in host immunity by enabling the cell to respond and interact with the environment around them, thus playing a critical role in intracellular signalling [43]. Therefore, enhancing any of these cells’ surface markers would suggest further anti-viral activity for SC234 and SC232.

J774.A.1 cells, when activated with LOX or LPS in the presence of SC234 and SC232, positively impact cell surface marker expression. LOX- and LPS-activated J774.A.1 in the presence of SC234 showed increases in CD80, CD86, CD40, MHCII, TLR2, and CD14. LOX-activated J774.A.1 in the presence of SC234 also showed increased MHCI, highlighting the specificity in bioactivity whereby SC234 has unique specificity in increasing the viral-associated cell surface marker MHCI, not seen in LPS-activated cells. LOX- and LPS-activated J774.A.1 in the presence of SC232 increase MHCII and TLR4. LOX-activated J774.A.1 cells in the presence of SC232 decrease CD14, CD40, and MHCI. Our results clearly demonstrate different effects on cells depending on either Lox or LPS activation.

MHCI plays a particularly important role in viral immunity for the detection of virally infected cytotoxic T lymphocytes [44]. CD80 and CD86 interact on APC and CD28 on T cells as costimulatory signals for the activation of T cells, are key players in anti-viral humoral and cellular immune responses, and play a critical role in the control of chronic and latent infections [45]. CD40 in particular is important for the restriction of infection of a broad range of RNA viruses and is critical for the control of RNA viruses over the first 24 h of infection [46].

Furthermore, TLR2 has been identified to play a role in viral immunity with protective roles against viruses such as varicella zoster virus, hepatitis C virus, vaccinia virus, cytomegalovirus, and respiratory syncytial virus [47].

Exopolysaccharides (EPS) from Cordyceps sinensis induces the expression of MHCII, CD40, CD80, and CD86 in DC sarcoma cells, enhances their ability of antigen uptake, and increases the secretion of IL-12 and TNF-α, thus suggesting that EPS have a critical role in initiating anti-tumour immunity and pro-inflammatory immune modulation [48]. Carrot pomace has also been found to increase the expression of co-stimulatory molecules CD40 and CD80, and the fraction of cells CD11c+MHCII+ cells in BMDCs increase pro-inflammatory cytokine production; in cyclophosphamide-immunosuppressed mice administered with influenza vaccine challenge, it significantly enhanced the efficacy of the influenza vaccine [49]. Resveratrol has been shown to enhance antigen presentation of peritoneal macrophages via the upregulation of CD86, MHCII, and TLR4 levels, suggesting its role as a pseudorabies virus vaccine-adjuvant therapy, aiding in the host protection against viral infection [50]. The similar effects of our fermentates on key cell surface markers involved in viral immunity further support their potential as anti-viral ingredients.

Having assessed and confirmed the immuno-supportive roles of SC234 and SC232 as immune-boosting compounds for defence against viral infection, we also assessed the effects of these novel fermentates on phagocytosis. Phagocytosis is another critical function of the macrophage in the defence against viral infection. In setting up the model for phagocytosis, the MFI of the latex beads and the percentage of phagocytosing cells within the population were measured. These two parameters were then combined in order to form the overall mean MPI, the combined effect of the MFI and percentage of phagocytosing cells, for a collective outlook of sample effect on phagocytosis.

In contrast to the viral immune-supportive roles identified for SC234 and SC232 so far, these fermentates negatively impact on the MPI. There is a small decrease in the MPI, meaning that the macrophage’s ability to phagocytose is negatively impacted in the presence of SC234 and SC232. This negative impact on phagocytosis is something to consider if these novel fermentates are to be considered for commercial development. However, it must be noted that the MPI for SC234 and R003 is enhanced in comparison to the RSM. This means that in comparison to the fermentate starting substrate, these novel fermentates in fact increase the ability of the macrophage to phagocytose. Phagocytosis is closely associated with bacterial and fungal clearance [51]. This critical role of phagocytosis in bacterial and fungal clearance highlights the importance of phagocytosis in the context of bacterial and fungal infections as opposed to viral infections, thus suggesting that in the context of viral immunity the role of phagocytosis may not be deemed as important as in the context of bacterial or fungal infection. Therefore, the decrease seen in the MPI as a result of the presence of fermentates SC234 and SC232 may still support the role of such novel fermentates in the context of viral immunity, as they provide an increase in MPI above the RSM non-fermented control.

It is clear from the literature that increases in phagocytotic activity through the use of functional foods can be linked with enhanced immunity. When RAW 264.7 murine macrophages are treated with wild simulated ginseng, increased phagocytotic activity is observed [52]. Panax ginseng Meyer, when administered to BALB/c mice, enhanced innate and adaptive immunity via the improved cell-mediated and humoral immunity, macrophage phagocytosis capacity, and NK cell activity [53]. In that study, He et al. hypothesised that the increased immunomodulating activity was due to the increased macrophage phagocytosis capacity, along with increased NK cell activity, enhancement of T and Th cells, as well as IL-2, IL-6, and IL-12 secretion and IgA, IgG1, and IgG2b production [53]. Fermenting *C. militaris* with *Pediococcus pentosaceus* ON89A (GRC-ON89A) can enhance phagocytosis in RAW 264.7 cells and primary peritoneal macrophages from normal mice and cyclophosphamide-immunosuppressed mice via the activation of MAPK and Lyn pathways [54]. It is suggested that GRC-ON89A has the potential to act as an immunostimulant for use as an immune-boosting therapy in immunosuppressed patients [54]. Whilst our findings demonstrate that SC234 and SC232 could impact positively on viral immune response, this study assessed this in comparison to any activity the RSM would have alone. This was in order to assess any advantage the fermentation of the RSM had in terms of bioactivity. A further comparison to other types of non-fermented substances will provide more information on how potent these fermentates are in supporting viral immunity.

## 5. Conclusions

As demonstrated from the current literature available on similar functional foods, we suggest a role for fermentates SC232 and SC234 as potential novel food ingredients for defence against viral infection in humans due to their overall positive effect on systemic immune responses. This is due to their ability to support the secretion of pro-inflammatory cytokines IL-1β, IL-6, TNF-α, IL-12p40, and IL-27 while increasing the anti-inflammatory cytokine IL-10 to maintain immune homeostasis, as well as via their NO induction to support the proliferation of the M1 pro-inflammatory phenotype for viral immunity. Overall, the samples’ ability to largely maintain chemokine expression, and in the case of MCP where this expression can be decreased, suggests a potential use of SC232 and SC234 as novel anti-viral and immune-supporting therapies. It is clear that increasing cell surface marker expression has a range of positive effects on a cell that can aid in adjuvant vaccine therapy, anti-tumour therapy, and immune-stimulating properties for overall immune boosting results for its host. Furthermore, functional food components which have the ability to enhance phagocytosis, like SC234 and SC232, above that of their starting substrate may have the potential to aid in boosting the immune system to provide enhanced innate and adaptive immunity, acting as potential immune-boosting therapies. However, it is important to consider the rate to which the overall phagocytosis is affected before consideration for commercial use to ensure the host is not negatively impacted. Therefore, we suggest the deeply impactful potential that our novel fermentates SC234 and SC232 have for defence against viral infection in humans.

## Figures and Tables

**Figure 1 nutrients-16-01212-f001:**
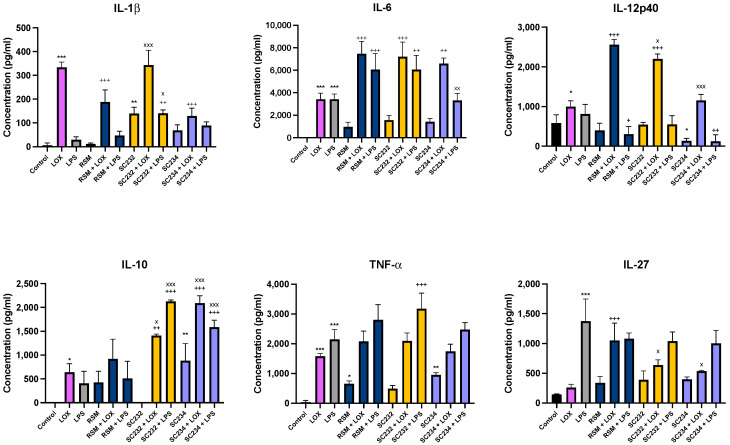
**Exposure of LOX- and LPS-activated BMDM to 25 mg/mL fermentates results in the secretion of cytokines.** BMDM cells were seeded at 1 × 10^6^ cells/mL and incubated overnight at 37 °C in 5% CO_2._ The following day, cells were stimulated with 25 mg/mL fermentate, incubated for 1 h at 37 °C in 5% CO_2_ and subsequently exposed to LOX 0.5 mM; LPS 100 ng/mL before incubating overnight under the same conditions. Non-fermented RSM was the fermentate control. Supernatants were removed after 24 h and ELISA was performed for cytokines IL-1β, IL-6, IL-10, TNF-α, IL-12p40, and IL-27. Data are presented as mean ± SEM of three replicates. Significance determined using one-way ANOVA with a Newman–Keuls post-test. Output *p* value style APA: 0.12 nonsignificant (unlabelled), 0.033 somewhat significant (*), 0.002 significant (**), and <0.001 highly significant (***); where the following symbols represent; (1) comparing control cells to LOX and LPS and unstimulated samples “*”, (2) comparing TLR to sample + TLR “+”, and (3) comparing RSM +/− TLR to sample +/− TLR “x”.

**Figure 2 nutrients-16-01212-f002:**
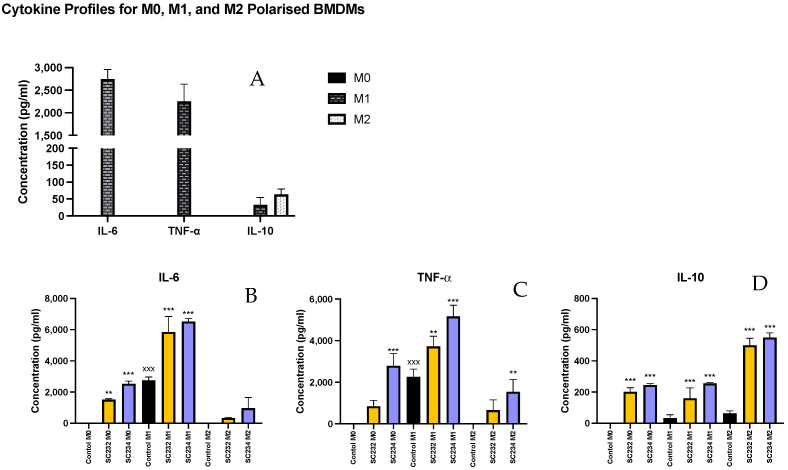
Exposure of M1/M2-polarised BMDMs to 25 mg/mL fermentates results in the secretion of cytokines IL-6, TNF-α, and IL-10. BMDM cells were seeded at 5 × 10^5^ cells/mL and incubated for 1 h at 37 °C in 5% CO_2._ Cells were stimulated with 25 mg/mL fermentates, incubated for 3 h at 37 °C in 5% CO_2_. The cells were either polarised to the M1 phenotype by stimulating with LPS (100 ng/mL) in the presence of 20 ng/mL rIFN-γ or towards M2 cells by adding 20 ng/mL rIL-4, 20 ng/mL IL-13, and 20 ng/mL rTGF-B and incubating for 24 h at 37 °C. Supernatants were removed after 24 h and ELISA was performed for IL-6, IL-10, and TNF-α. Data are presented as mean ± SEM of three replicates. (**A**) represent the M0, M1, and M2 profiles for each cytokine. (**B**–**D**) represent cytokine output in response to sample presence. Significance determined using one-way ANOVA with a Newman–Keuls post-test. Output *p* value style APA: 0.12 nonsignificant (unlabelled), 0.033 somewhat significant (*), 0.002 significant (**), and <0.001 highly significant (***); where the following symbols represent; (1) comparing fermentates to polarised control cells “*”, and (2) comparing M0 to M1 and M2 controls “x”.

**Figure 3 nutrients-16-01212-f003:**
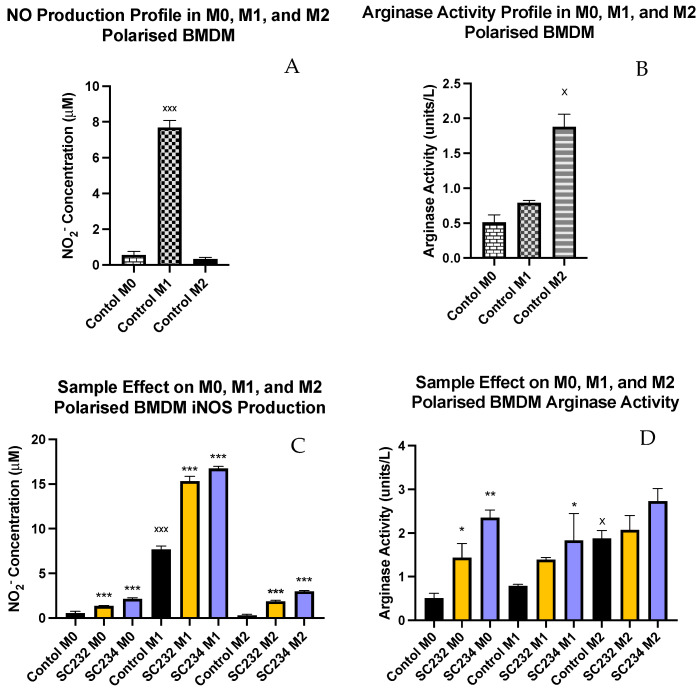
**Exposure of M0, M1, and M2 BMDMs to 25 mg/mL fermentates affects production of NO production and arginase activity.** BMDM cells were seeded at 5 × 10^5^ cells/mL and incubated for 1 h at 37 °C in 5% CO_2._ Cells were stimulated with 25 mg/mL fermentates and incubated for 3 h at 37 °C in 5% CO_2._ The cells were either polarised to the M1 phenotype by stimulating with LPS (100 ng/mL) in the presence of 20 ng/mL rIFN-γ or towards M2 cells by adding 20 ng/mL rIL-4, 20 ng/mL IL-13, and 20 ng/mL rTGF-B and incubating for 24 h at 37 °C. Supernatants were removed after 24 h and Griess assay was performed as per manufacturer’s instructions for determination of NO production (**A**,**C**). Cell lysates were prepared, and arginase assay carried out to determine arginase activity (**B**,**D**). Data are presented as mean ± SEM of three replicates. (**A**,**B**) represent the NO and arginase activity profiles for M0-, M1-, and M2-polarised cells, respectively. Significance determined using one-way ANOVA with a Newman–Keuls post-test. Output *p* value style APA: 0.12 nonsignificant (unlabelled), 0.033 somewhat significant (*), 0.002 significant (**), and <0.001 highly significant (***); where the following symbols represent; (1) comparing fermentates to polarised control cells “*” and (2) comparing M0 to M1 and M2 controls “x”.

**Figure 4 nutrients-16-01212-f004:**
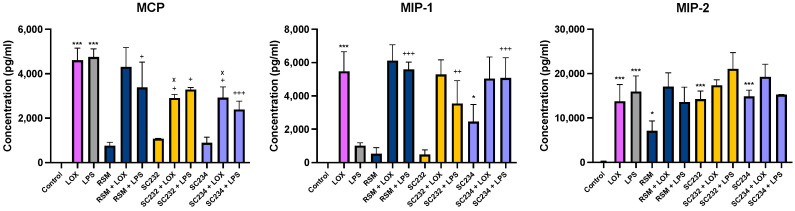
**Exposure of LOX- and LPS-activated BMDMs to 25 mg/mL fermentates results in the secretion of chemokines.** BMDM cells were seeded at 1 × 10^6^ cells/mL and left overnight at 37 °C in 5% CO_2._ The following day, cells were stimulated with 25 mg/mL raw sample fermentate, incubated for 1 h at 37 °C in 5% CO_2,_ and subsequently exposed to LOX 0.5 mM; LPS 100 ng/mL before incubating overnight under the same conditions. RSM was the fermentate control. Supernatants were removed after 24 h and ELISA was performed for MCP, MIP-1, and MIP-2. Data are presented as mean ± SEM of three replicates. Significance determined using one-way ANOVA with a Newman–Keuls post-test. Output *p* value style APA: 0.12 nonsignificant (unlabelled), 0.033 somewhat significant (*), 0.002 significant (**), and <0.001 highly significant (***); where the following symbols represent; (1) comparing control cells to LOX and LPS, and unstimulated samples”*”, (2) comparing TLR to sample + TLR “+”, and (3) comparing RSM +/− TLR to sample +/− TLR “x”.

**Figure 5 nutrients-16-01212-f005:**
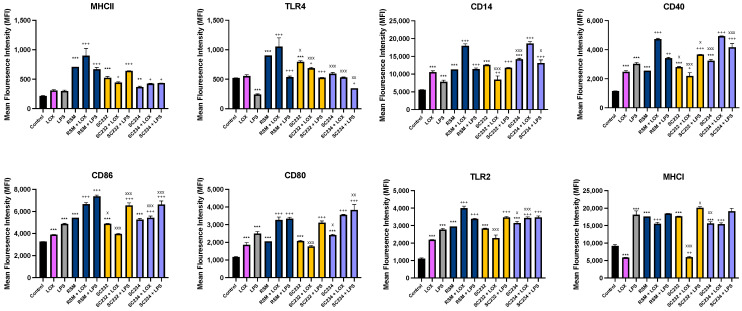
**Exposure of LOX- and LPS-activated J774.A.1 to 25 mg/mL fermentates affect cell surface marker expression.** J774.A.1 cells were seeded at 1 × 10^6^ cells/mL and incubated overnight at 37 °C in 5% CO_2._ After 24 h, cells were stimulated with 25 mg/mL fermentates and incubated for 1 h at 37 °C in 5% CO_2_ before stimulating with LOX 0.5 mM or LPS 100 ng/mL. Cell suspensions were retained, and cell-staining protocol was carried out to assess the presence of cell surface markers MHCII, TLR4, CD86, CD80, CD14, CD40, TLR2, and MHCI in the presence of fermentate sample. Cells were analysed using a BD FACSAria 1 system flow cytometer, raw FCS files analysed, and data graphed using V10.0 FlowJo software. Data are presented as mean ± SEM of two replicates. Significance was determined using one-way ANOVA with a Newman–Keuls post-test. Output *p* value style APA: 0.12 nonsignificant (unlabelled), 0.033 somewhat significant (*), 0.002 significant (**), and <0.001 highly significant (***); where the following symbols represent; (1) comparing control cells to TLR controls and unstimulated samples “*”, (2) comparing TLR controls to sample + TLR “+”, and (3) comparing RSM +/− TLR to sample +/− TLR “x”.

**Figure 6 nutrients-16-01212-f006:**
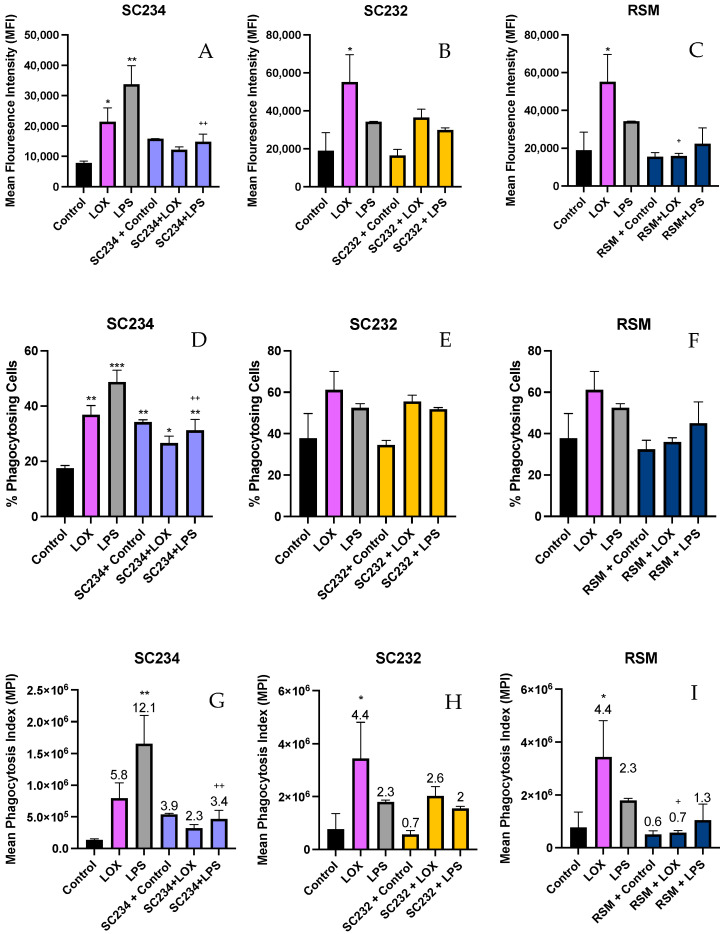
**Exposure of LOX- and LPS-activated J774.A.1 to 25 mg/mL fermentates affects MFI, % phagocytosing cells, and MPI**. J774.A.1. cells were seeded at 1 × 10^6^ cells/mL and incubated overnight at 37 °C in 5% CO_2._ The following day, cells were stimulated with 25 mg/mL fermentates for 1 h before activating with LPS 100 ng/mL and LOX 0.5 mM, and incubated for 4 h at 37 °C in 5% CO_2._ Cell suspensions were retained, and cells were stimulated with 1 µm fluorescent latex beads at a concentration of 20 beads per cell for 1 h at 37 °C in 5% CO_2._ Cells were analysed using a BD FACSAria 1 system flow cytometer, raw FCS files were analysed, and data graphed using V10.0 FlowJo software. Data are presented as mean ± SEM of two replicates. (**A**–**C**) represent MFI, (**D**–**F**) represent percentage of phagocytes, and (**G**–**I**) represent MPI. Significance determined using one-way ANOVA with a Newman–Keuls post-test. Output *p* value style APA0.12 nonsignificant (unlabelled), 0.033 somewhat significant (*), 0.002 significant (**), and <0.001 highly significant (***); where the following symbols represent; (1) comparing control cells to each test cell “*”, (2) comparing each corresponding sample + TLR to TLR alone “+”. MPI data analysed as a product of the percentage phagocytosing cells and MFI. Number indicated above bar is the MPI compared to the control cells, represented as 1.

## Data Availability

The raw data supporting the conclusions of this article will be made available by the authors on request.

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
