# Peer review of "Novel Fermentates Can Enhance Key Immune Responses Associated with Viral Immunity"

_nutrients, 2024, doi:10.3390/nu16081212_

Round 1
Reviewer 1 Report
Comments and Suggestions for Authors
The manuscript by Dearbhla Finnegan aimed to investigate the effect of fermentates derived from the lactic acid bacteria of dairy products on host immune responses. Their data showed that the Lb. helveticus strains SC234 and SC232 can increase the production and secretion of cytokines, such as IL-1β, IL-6, TNF-α, and IL-10, as well as promote an M1 proinflammatory phenotype for viral immunity via NO induction. Meanwhile the author also found SC234 and SC232 can decrease chemokine expression of MCP, increase cell surface marker expression, and enhance phagocytosis in comparison to their starting material. These data could lay a foundation that using these novel fermentates as novel functional food ingredients for the treatment, management and control of viral infection. However, the data presentation in this paper needs to be improved. The paper should be written more clearly.
1. The authors should label correctly in the figures. Even some figures lack labels.
2. I suggest the authors shorten the discussion, making clearer for reader’s understanding.
Author Response
We would like to thank the reviewers for their time in assessing our manuscript. We appreciate your comments and have done our best in addressing each recommendation.
Reviewer 1
Comment 1:
“The authors should label correctly in the figures. Even some figures lack labels.”
Response:
Thank you for this comment. This error has been fixed and we believe this to have been a formatting issue that happened. Figures are now correctly labelled and located.
Comment 2:
“I suggest the authors shorten the discussion, making clearer for reader’s understanding.”
Response:
The discussion section has now been shortened as suggested by the reviewers.
Reviewer 2 Report
Comments and Suggestions for Authors
The manuscript titled "Novel fermentates can enhance key immune responses associated with viral immunity" investigates the immunomodulatory effects of novel milk fermentates derived from Lb. helveticus strains on murine macrophage models. Here's my evaluation highlighting potential flaws that once considered and replied can imprive the paper:
Major Criticism:
The controls used, such as non-fermented reconstituted skim milk (RSM), were effective, but the study could enhance its robustness by including additional controls, such as other non-fermentative substrates, to confirm that the observed effects are unique to the specific fermentates tested.
Further
While the study uses a single concentration of fermentates, a dose-response analysis could provide more insights into the potency and efficacy of these fermentates, helping to establish a clearer therapeutic index.
Finally, the paper focuses on specific markers of macrophage activation but does not address whether these changes lead to an overall beneficial or potentially harmful systemic immune response. Inclusion of studies on the effect of these fermentates on other immune cells or in a whole organism could provide a more comprehensive understanding of their immunological impact. If the authors have the possibility to add information also on other cell subset it would greatly improve the paper
Please Provide this information or address as you can to that consideration
Reviewer 3 Report
Comments and Suggestions for Authors
This is an interesting article by Dearbhla Finnegan et al.
I would like to address a small number of suggestions to you.
General recommendations
Page 1, line 41
Please start to write the name of lactobacilli and bifidobacteria with capital letter.
Page 1, line 43
(Pressione, 2012)? Is this a reference of article by Enrica Pressione entitled "Lactic acid bacteria contribution to gut microbiota complexity: lights and shadows Enrica Pressione" at Front. Cell. Infect. Microbiol., 2012?
Page 2, line 51
Please correct the name of SARS CoV-2 viruse.
Page 2, line 60
Please correct "IL (interleukin)-1β" to interleukin-1β (IL-1β)
Page 4, lines 177-183
The starting of results with "data not shown" overshadows all your results.
Please rewrite the paragraph "An MTS assay confirmed that the fermentates SC232 and SC234 in the presence/absence of LOX or LPS had no effect on the viability of either J774.A.1 cells and BMDMs (data not shown). Initially, an ELISA was performed on the cell line J774.A.1 macrophage to assess bioactivity of fermentates SC232 and SC234 in the presence/absence of LOX or LPS. The novel fermentates altered the secretion of cytokines in response to LOX and LPS when compared to the respective controls in J774.A.1s (data not shown)."
Page 6, figure 2
The title "Cytokine Profiles for M0, M1, and M2 Polarised BMDMs" must be deleted.
Figure 3
The figure 3 needs to be removed. This figure is presented at 2 pages (page 7 and 8 as two different subfigures.
Please keep on the text only figure 3 at page 9
Figure 6 demonstrates the phagocytosis after stimulation of J774 cells with LOX and LPS, in the presence of SC234 or SC232. The titles such as "SC234", confuse the readers. You run the first panel "A", with or without SC234. First three bars, black, pink, and gray were indicated in the figure as control, LOX and LPS. Unfortunately, it is not clear if next bar was control with SC234.
Page 14, line410
The sentence “Macrophage play a critical role in viral immunity” must be rephrased because the macrophages play a key role in the innate immune, not only in viral immunity.
Round 2
Reviewer 2 Report
Comments and Suggestions for Authors
The modifications made to the paper slightly improve the original text without achieving the purpose of the comments. For example, in the text the authors write, "Initially preliminary studies carried out by the laboratory on a large panel of fermentates used a dose range of 5mg/ml, 10mg/ml, 25mg/ml, 50mg/ml fermentates to reveal 25mg/ml as the optimal dose for fermentate bioactivity, and thus this dose of 25mg/ml was used for such further in-depth analysis." However, they do not cite sources nor specify "unpublished data." The discussion should be revised more accurately according to the previous suggestions.
Author Response
We would like to thank the reviewer for their additional comments. We appreciate your time in contributing to bettering our manuscript.
In relation to the addition of unpublished data, we have subsequently added this to the following sentence: "Initially preliminary studies carried out by the laboratory on a large panel of fermentates used a dose range of 5mg/ml, 10mg/ml, 25mg/ml, 50mg/ml fermentates to reveal 25mg/ml as the optimal dose for fermentate bioactivity, and thus this dose of 25mg/ml was used for such further in depth analysis (unpublished data).". We have been asked to remove other references of unpublished work from other reviewers as they felt it overshadowed the results, however we believe this to be the appropriate way to include both comments from all reviewers.
Additionally, in order to increase the accuracy of our discussion as suggested by the reviewer we have included reference to non-fermentative substances and how these would enhance the study providing additional information on the potency of these fermentates as follows: "Whilst our findings demonstrate that SC234 and SC232 could impact positively on viral immune response, the study assessed this in comparison to any activity the RSM would have alone. This was in order to assess any advantage the fermentation of the RSM had in terms of bioactivity. A further comparison to other types of non-fermented substances will provide more information on how potent these fermentates are in supporting viral immunity.".